# Seasonal and Scale Effects of Anthropogenic Pressures on Water Quality and Ecological Integrity: A Study in the Sabor River Basin (NE Portugal) Using Partial Least Squares-Path Modeling

António Carlos Pinheiro Fernandes [1], Luís Filipe Sanches Fernandes [1], Daniela Patrícia Salgado Terêncio [1], Rui Manuel Vitor Cortes [1] and Fernando António Leal Pacheco [2,*]

[1] Centro de Investigação e Tecnologias Agroambientais e Biológicas, Universidade de Trás-os-Montes e Alto Douro, Ap 1013, 5001-801 Vila Real, Portugal; acpf91@utad.pt (A.C.P.F.); lfilipe@utad.pt (L.F.S.F.); dterencio@utad.pt (D.P.S.T.); rcortes@utad.pt (R.M.V.C.)

[2] Centro de Química de Vila Real, Universidade de Trás-os-Montes e Alto Douro, Ap 1013, 5001-801 Vila Real, Portugal

* Correspondence: fpacheco@utad.pt

**Abstract:** Interactions between pollution sources, water contamination, and ecological integrity are complex phenomena and hard to access. To comprehend this subject of study, it is crucial to use advanced statistical tools, which can unveil cause-effect relationships between pressure from surface waters, released contaminants, and damage to the ecological status. In this study, two partial least squares-path models (PLS-PM) were created and analyzed in order to understand how the cause-effect relationships can change over two seasons (summer and winter) and how the used scale (short or long) can affect the results. During the summer of 2016 and winter of 2017 surface water parameters and the North Invertebrate Portuguese Index were measured in strategic sampling sites. For each site, it two sections were delineated: the total upstream drainage area (long scale) and 250 m (short scale). For each section, data of pressures in surface waters including point source, diffuse emissions and landscape metrics were gathered. The methodology was applied to the Sabor River Basin, located in the northeast of Portugal. In this study, it was possible to determine in which season pressures affect ecological integrity and also which scale should be addressed. The models showed the influences of manganese and of potassium concentrations in stream water on the decrease in summer water quality, while arsenic's harmful effect occurs during winter. Pastures and environmental land use conflicts were considered threats to water quality when analyzed on a long scale, whereas agricultural areas played a role when the short scale was used. The effect of landscape edge density revealed to be independent of scale or season. Effluent discharges in surface water affected the water quality during the summer season, while the effect of discharges in groundwater affected the water quality in winter. It has also been found that, to find the harmful effect of pressures, it is necessary to approach different scales and that the role of landscape metrics can also overlap contaminant sources.

**Keywords:** water quality; pollution sources; landscape metrics; scale effects; seasonality; PLS-PM

## 1. Introduction

Stream water is a part of the hydrosphere that includes many interactions between natural and anthropogenic effects, namely rock weathering and water contamination, which sometimes overlap or even interfere with each other [1–7]. With social and demographic expansion, many threats

to water quality have arisen and reached worrying levels around the globe. Many studies have made inventories of pollution causes and have quantified the roles of pollutants at a catchment scale [8–11]. Effluent discharge is the most unsettling point source pressure that leads water to degrade [12]. Discharge of urban effluents is known to contain higher organic loads, comparatively to industrial effluents [13]. The composition of industrial discharge is highly variable based on the type of industry and degree of treatment. Typically, high loads of heavy metals are released from industrial activities such as mining [14] or metallurgic activities [15]. However, even in industrial effluents, it is possible to find high concentrations of organic compounds, such as from industries related to food production [16,17]. Livestock production [18] and agriculture [19] exert diffuse threats on water quality, which are mostly associated with the transport of pesticides and nutrients through infiltration into groundwater and runoff into rivers. The consequences of these pressures can be moderated by the environment during the course of processes such as phytoremediation [20] or natural aeration induced by turbulence [21], which promotes the self-depuration capacity of rivers [22]. Riparian vegetation has high potential to improve and sustain water quality because it comprises a vegetation barrier against the propagation of contaminants [23] and promotes a natural environment for aquatic lifeforms, thereby improving biodiversity [24]. The benefits of riparian vegetation can be accessed through landscape metrics [25], such as connectance, density, and contrast (e.g., [26,27]). For other types of land use patches, these metrics reveal an intrinsic relationship with water quality [28,29]. For this reason, several authors have used landscape metrics to explain the degradation of water quality by establishing relationships with different types of contaminants [30,31]. When such interactions are studied, it must be accessed a suitable spatial extent for statistical sampling [32]. In general, the scale can be comprised of buffers, riparian extents, and catchments [33,34]. By comparing studies that accessed different types of scales, is found some inconsistency in the results, as some authors argue that the use of whole watersheds is more feasible [32,35] while others report that the riparian scale is the more appropriate [25,28,36]. These conflicts in opinion are due to variations in study design, study areas, topographic aspects [37] and, in particular, the season [38] at which measurements were made.

Another relevant issue in water quality assessments is the relationship between land use and land capability (natural use). When actual land uses differ from land capability, an environmental land use conflict develops [39]. This analysis settles on the principle that agricultural areas and livestock production can place heavy demands on soils, while forested areas conserve and protect the soil [39,40], looking forward to a sustainable land use policy [41]. Many authors have studied the consequences of land use conflicts, such as the increase in flood vulnerability [42], decrease in soil fertility [40], and the increase of soil erosion [43,44]. Land use conflicts can be a significant threat to water quality and ecological integrity because they tend to amplify harmful effects on groundwater [45,46], surface water [47], and the biodiversity of riverine ecosystems [48] in rural catchments dominated by agriculture and pasture.

One of the most common methods to reveal the effects of pollution is by measuring contaminant concentrations in surface waters. By comparing measurements from previous periods, or even, with legislative limits, it is possible to determine if these are looming concentrations. The point is that damage is directly felt in lifeforms living in surface waters and can occur even when concentrations are within legal limits [49,50]. For such reasons, ecological integrity should be encompassed in environmental studies, which is the ecosystems ability to persist when natural or anthropogenic changes occur [51,52]. When pressures on freshwater resources are excessive, damage to ecological integrity can be accessed through bio-indicators such as IBMWP [53], $IPtI_N$ [54] or MELI [55].

To fully understand the interactions between pollution sources, water contamination, and ecological integrity, it is worthwhile to use statistical models such as eigenvector based models [56,57] or structural equation models (SEM). The most common types of SEM are CB-SEM (covariance-based SEM) and SEM-PLS (SEM-Partial Least Squares), also termed PLS-PM (PLS-Path Modeling) [58]. In the first case, the estimation of model parameters is based on maximum likelihood, while for PLS-PM, the estimation procedure resorts to ordinary least squares regressions [59]. SEM

models have been widely used in the social sciences, but nowadays, the method applies to other areas of scientific knowledge. One reason for this expansion is due to the current need to address complex interactions among a large number of parameters. In previous applications of SEM to water quality assessments, some interactions between surface water parameters have been exposed, but the specific roles of anthropogenic pressures and natural processes have not been quantifies [60]. In a study addressing groundwater quality, SEM successfully explained the chemical composition of phreatic aquifer groundwater samples and the origin of water mineralization [61]. In the Feitsui Reservoir Watershed, SEM was used as a mathematical tool for the evaluation of harmful effects resulting from three types of pollution: organic, sediment, and eutrophication [62]. From a social standpoint, SEM was used to understand citizen's awareness about water quality [62,63].

A recent application of SEM explored interactions between anthropogenic pressures, water contamination, and ecological integrity in two very different river basins, one urbanized and densely populated and another rural and sparsely populated [64]. Following that work, the present study aimed to discover seasonal and scale effects potentially affecting the aforementioned interactions using PLS-PM, because those effects can be substantial and have not yet been investigated. This study presents itself as an innovative analysis since there are few environmental studies using structural equation models. Another breakthrough aspect is that the effects of landscape metrics are studied in conjunction with contaminant emissions. These two types of variables were included in the same model, accessing the interplay with ecological integrity, along with impact changes across different scales and seasons.

## 2. Methodology

### 2.1. Study Area

The Sabor River Basin is an Iberian watercourse and one of the largest tributaries of the Douro River [65]. It belongs to Hydrographic Region 3 (HR 3) [66] and is located in the northeast part of Portugal (Figure 1). The watershed covers an area of approximately 3297 km$^2$. A large portion (83%) of this catchment is located in Portugal, while a small portion (17%) occupies Spanish territory [66]. According to the Portuguese map of land uses [67] 35% of the area is occupied by agriculture, 62% by forest, 2% by artificial surfaces, and 1% by water bodies. Within the Sabor catchment, the population density ranges from 9.7 to 30.1 inhabitants/km$^2$ [64], which is not very high when compared to other regions in Portugal. Parishes that belong to Sabor River Basin are predominantly rural, due to the low population density. In this location, there is high economic dependence on rural activities such as livestock production and agriculture [68,69]. The climate in Sabor River Basin is Mediterranean. The average temperature ranges from 1 °C in winter to 27.6 °C in summer [70], and rainfall ranges from 480 to 1360 mm/year [71]. Data on the ecological status indicate that approximately 80% of all Sabor streams and streamlets are in good condition [66]. Comparatively to other river basins such as Ave, Sabor River Basin has minimal pollution, since in Ave more than 70% of the streams are under a poor ecological status [72]. The application of environmental studies in polluted river basins is mandatory since it can be prioritized in which pressures should be taken actions, looking forward to improving quality. Although the research should be applied for mitigation or recovery purposes, it is also necessary to study basins with overall acceptable status, in order to prevent decay, or even if possible, to improve.

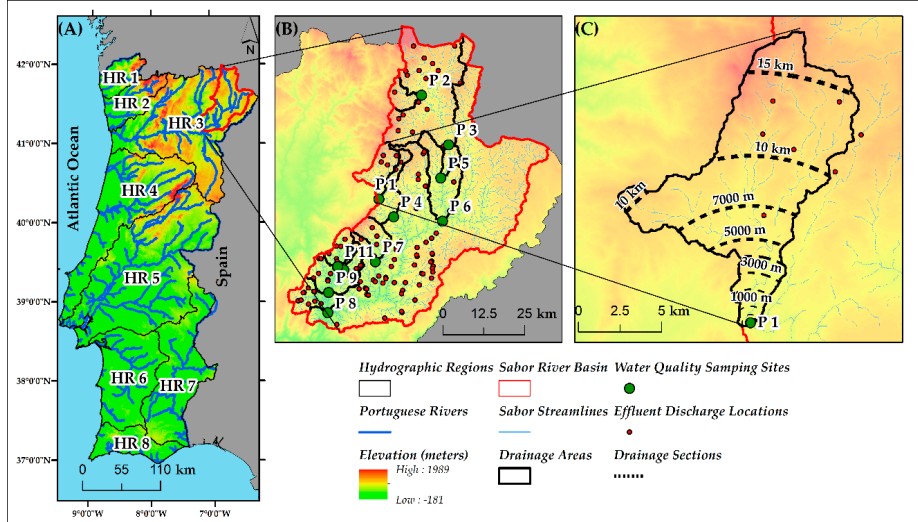

**Figure 1.** (**A**) Distribution of hydrographic regions in Portugal. (**B**) Sabor River Basin, water quality sampling sites, and effluent discharge sites. (**C**) Drainage area of sampling site P1 and drainage sections.

## 2.2. Materials and Methods

Figure 2 portrays the methodological workflow. In a first step, data on environmental parameters were downloaded from official sources and summarized in a dataset called the Sabor River Basin dataset. These data comprised information on anthropogenic pressures (contaminant sources [73]), surface water parameters (SWP), and ecological integrity. Table 1 the data sources. The second step defined a number of sampling sites and drew 250 m buffers around them. It also delineated the drainage areas upstream of the sampling sites. Figure 1B shows the sampling site locations and Figure 1C shows the drainage area of site P1. The second step continued with estimation of an average value for each pressure within the buffers and the drainage areas, while 24 surface water parameters (listed in Table 1) and a biological index (North Invertebrate Portuguese Index—IPtI$_N$) were evaluated at the sampling site in the summer of 2016 and winter of 2017. In a third step, two PLS-PM models were developed. The first model, called the short-scale model, used data from the 250 m buffers. The second model, called the long-scale model, involved data from the entire drainage area. The aim was to compare relationships among pressures, water quality, and ecological integrity across the two scales. The third step comprised the division of both models into summer and winter sub-models. The aim was to compare the pressures, water quality, and ecological integrity relationships across the two seasons.

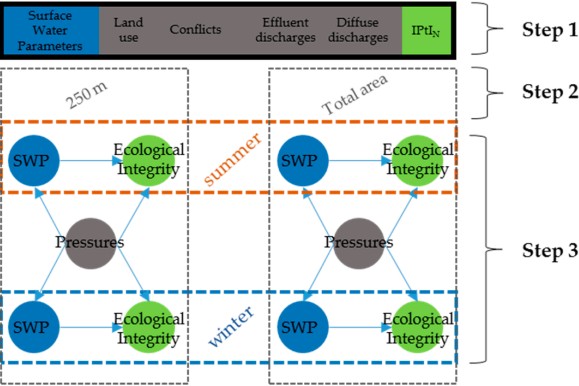

**Figure 2.** Methodological workflow. Pressures refer to land uses and environmental land use conflicts as well as point source and diffuse effluent discharges. SWP refers to the surface water parameters identified in Table 1. The assessment of ecological integrity was based on the IPtI$_N$ index.

**Table 1.** Sources of information used in the PLS-PM (PLS-Path Modeling) model.

| Data | Description | Source |
| --- | --- | --- |
| Elevation model | Elevation model raster file with a pixel size of 25 × 25 m. | [74] |
| Point sources of contamination | Yields of discharged biological and chemical oxygen demands, nitrogen and phosphorus from urban effluents in surface water (Effluents (hydric)) and soil/underground water (Effluents (soil)). | [73] |
| Diffuse sources of contamination | Nitrogen and phosphorous yields sourced from agriculture plus forested areas (FA N) and from livestock production areas (LS N). | [73] |
| Land use | Land use map of Portuguese territory—CLC2015 (reference: 2015) and metrics derived therefrom (see description in text): "area used for agriculture", "npc", "edge density", "ci cp" and "ci pp". | [67] |
| Land use conflicts | Difference between land occupation and land capability. | [9] |
| SWP | Values of pH, temperature, conductivity, dissolved oxygen, nitrites, nitrates, sulfates, phosphates, total suspended solids, calcium, iron, magnesium, potassium, sodium, total aluminum, arsenic, cadmium, lead, cobalt, copper, manganese, zinc, nickel, and chromium. | Measured in the field during the summer of 2016 (S 16) and winter of 2017 (W17) |
| $IPtI_N$ | Biodiversity of benthic macroinvertebrates | Measured in the field in the summer of 2016 (S 16) and winter of 2017 (W17) |

## 2.3. Dataset Preparation

The coordinates of sampling sites were used to create point shapefiles in ArcGIS [75]. Environmental applications of this software are widespread in the scientific literature [49,76–82]. An elevation raster file [74] and ArcHydro tools [83] were used to delineate drainage areas and streamlines. For each drainage area, the effluent discharge was evaluated for each contaminant at each discharge point (red circles in Figure 1B). The effluent discharge was divided by the logarithm distance between the discharge point and the sampling point (P in Figure 1B), and then summed for nitrogen, phosphorous, and biological and chemical oxygen demands (BOD and COD) using the Spatial Join ArcMap tool to identify which discharge points were inside the drainage areas. Then, the Pivot Table tool was used to calculate the sum of each effluent discharge divided by the logarithm distance. It also calculated the amount of N and P from agriculture, forest, and livestock production in the 250 m buffers and drainage areas using the Zonal Statistics as Table ArcMap tool.

The environmental land use conflicts were also evaluated within these domains. The method used is described in Appendix A. Landscape metrics were calculated using a specific toolbox embedded in ArcGIS [27]. Metrics were calculated for the 250 m buffers and drainage areas using the COS 2015 [67] which is the most recent land use map of Portugal. They were also calculated for land use conflict patches.

The SWP and $IPtI_N$ were collected from 11 sampling sites (Figure 1B). The SWP measurement procedures were performed in accordance to the official methods, adopted under the EU Water Framework Directive 2000/60/EC (WFD) in Portugal, described in Appendix B. The $IPtI_N$ is proportional to the amount and diversity of benthic macroinvertebrates in freshwater, being quantified as a score between 0 and 1 [54]. High scores represent sites holding pollution sensitive organisms, while low scores are representative of polluted locations. Since benthic macroinvertebrates are vulnerable to all forms of pollution [84,85] this index is used to qualify the ecological status of stream water according to five classes: "Bad", "Poor", "Moderate", "Good", and "High" [54]. For a detailed explanation, see Appendix C.

The data were compiled in two Excel worksheets, one for each PLS-PM model. Due to collinearity problems assessed by variance inflation factors (VIF) [86], many variables had to be removed from the PLS-PM models. Finally, 23 variables were used to build the PLS-PM constructs. The $IPtI_N$ was used

to evaluate the ecological integrity, while the scores of arsenic, dissolved oxygen, manganese, nitrates, potassium, and pH were used to evaluate the water quality. The mass flow of nitrogen (divided by the logarithm of the distance between the discharge point and the sampling site) quantified the effluent discharge directly in surface water ("Effluents (hydric)") or in soil ("Effluents (soil)"). The number of landscape metrics used in the model was 5. The first metric was the total area related to agricultural land uses. The second was the number of patches (npc) related to severe environmental land use conflicts (class 3). The edge density of all land uses was the third metric, calculated by the sum of patches length, divided by the area. Within pastureland, the fourth metric was the connectivity between patches measured as the ratio (%) between the total area of neighbor patches (500 m distance) and the area of an envelope covering those patches (ci cp). The fifth metric was similar to the fourth metric, but it applied to water bodies, namely rivers, streams, streamlets, and lakes (ci pp).

All variables were grouped into 7 latent variables: "Diffuse Emissions", "Land Use", "Point Source Emissions", "SWP (S 16)", "SWP (W 17)", "Ecological Integrity (S 16)", and "Ecological Integrity (W 17)". Figure 2 illustrates the relationships among latent variables and bearing-measured variables as well as between latent variables. The computer program used to execute PLS-PM was SMART-PLS [87]. It was used to create 2 SEM-PLS models based on the same conceptual model described by other authors [64,88]. Pollution sources are connected to an increase in surface water parameters (in the form of contamination) and a decrease in ecological integrity, while surface water parameters decrease ecological integrity. Based on this perspective, three latent variables were created for each type of pollution source (also called as pressure):"Point Source Emissions" composed of the measured variables representing the discharge of effluents into surface water and soils; "Diffuse Emissions" containing the emissions from livestock, forest, and agriculture; and "Land Uses" containing the data from landscape metrics. For each model, the values of the referred measured variables (pressures) were different, because in model 1, the pressure values were taken from the sampling sites to an upstream distance of 250 m, while for model 2, the values were collected for the entire drainage area. The other latent variables were relative to the data of surface water parameters "SWP" and "Ecological Integrity" represented by $IPtI_N$. These latent variables were represented twice in each model, in order to contain data collected in the summer of 2016 (S 16) and winter of 2017 (W 17). The values of the measured variables that composed "SWP (S 16)", "SWP W 17)", "Ecological Integrity (S 16)", and "Ecological Integrity (W 17)" were equal for both model 1 and model 2, because the sampling sites were the same. Inside each model, it was possible to compare the seasonal effects for the same scale, since in each model, the "SWP" and "Ecological Integrity" data were represented twice, once for each season. When comparing the models, it was possible to access seasonal and scale effects simultaneously, because the input data for latent variables relative to pressures were different in each model. The used algorithm attributed weights to the measured variables and patch coefficients to the connections between latent variables in order to maximize the $R^2$ values. For a detailed explanation of the calculation procedure, please see the recent work by Terêncio et al. [89].

## 3. Results

### 3.1. Spatial Data and Water Quality Parameters

Figure 3 portrays the spatial data. The diffuse discharge provided by the Environmental Agency of Portugal do not cover the entire river basin, just the sampled drainage areas. All the point source emissions are only from urban origin, because industrial activity is barely present in the studied region. The highest concentration of discharge points occurs near to the Sabor River downstream area. According to the land cover map (Figure 3D), the predominant land uses are agriculture and forestry. The three classes of land use conflict are scattered across the river basin.

Table 2 depicts the concentration of water quality parameters and $IPtI_N$ index. The Portuguese requirements for drinking water were checked [90]. In that context, the pH should be between 6.5 and 8.5. For dissolved oxygen, there is no threshold to respect, but for nitrate, potassium, arsenic,

and manganese, the maximum recommended values are 25 mg/L, 10 mg/L, 50 µg/L, and 20 µg/L, respectively. Compared to WFD requirements, the criteria is stricter only for Arsenic, the maximum concentration being equal to 10 µg/L. In Table 2, the parameters that exceed the legally recommended thresholds are marked in orange. Overall, the pH values are within the required range for sites P3 and P5 in summer, while P4, P5, and P7 are slightly above the range. The summer concentrations of manganese are high, being significantly above the maximum recommended values. The ecological status varies between moderate and high in summer and winter.

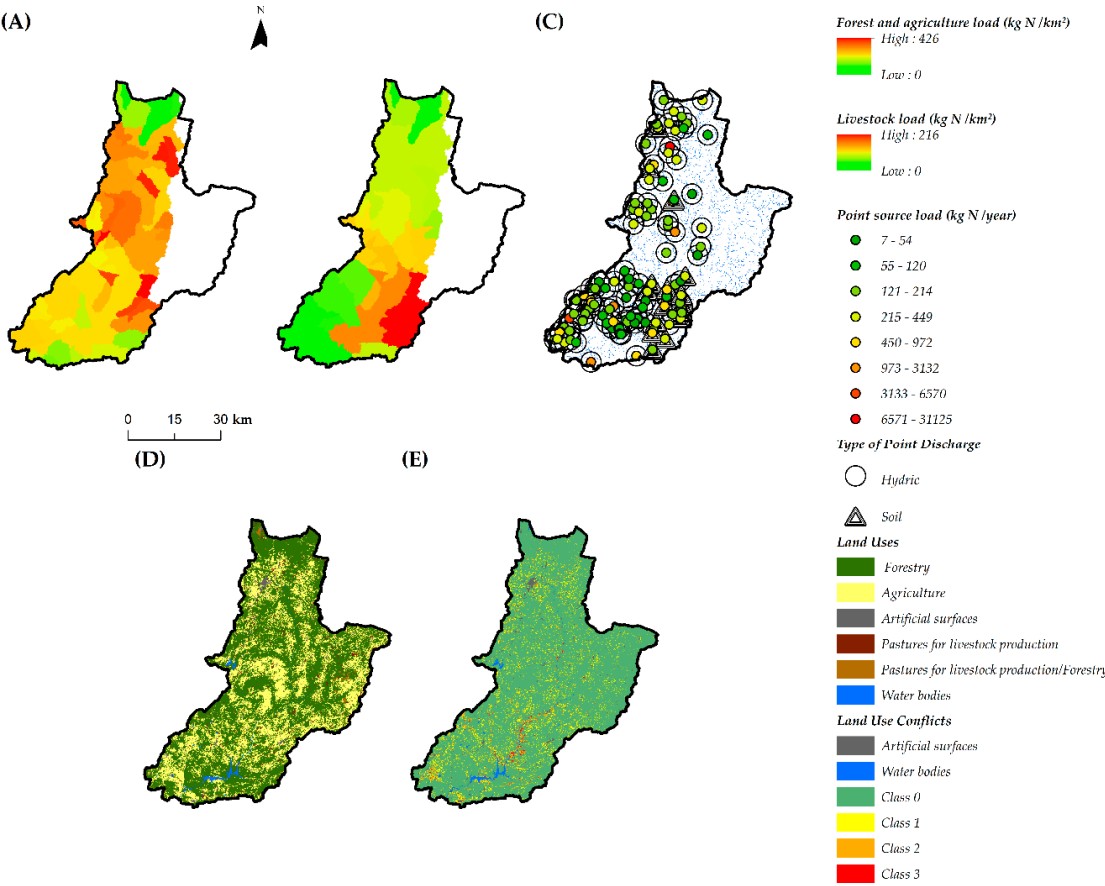

**Figure 3.** Spatial distribution of pressure variables used in the PLS-PM models: (**A**) forest and agriculture loads of nitrogen; (**B**) livestock loads of nitrogen; (**C**) point source discharges of nitrogen in soil and surface water; (**D**) land uses; (**E**) land use conflicts.

**Table 2.** Characterization of sampling sites surface water parameters and ecological integrity (IPtI$_N$). In orange are highlighted the exceedances above Portuguese legislation thresholds and WFD directive.

| | | IPtI$_N$ | | pH | Dissolved Oxygen mg (O$_2$)/L | Nitrates mg (NO$_3$)/L | Potassium mg (K)/L | Arsenic µg (As)/L | Manganese µg (Mn)/L |
|---|---|---|---|---|---|---|---|---|---|
| Summer 2016 | P 1 | 0.83 | Good | 7.95 | 8.55 | 8.49 | 0.97 | 2.25 | 9.03 |
| | P 2 | 0.62 | Moderate | 7.73 | 7.19 | 15.05 | 4.05 | 2.31 | 53.79 |
| | P 3 | 0.72 | Good | 8.78 | 10.23 | 3.80 | 1.60 | 6.33 | 132.10 |
| | P 4 | 0.89 | Excellent | 8.13 | 8.27 | 2.28 | 0.52 | 2.38 | 45.80 |
| | P 5 | 0.84 | Good | 8.71 | 4.03 | 1.52 | 1.19 | 0.03 | 101.00 |
| | P 6 | 0.51 | Moderate | 7.37 | 6.88 | 2.00 | 1.16 | 0.03 | 60.66 |
| | P 7 | 0.78 | Good | 6.95 | 7.56 | 1.59 | 0.51 | 12.92 | 88.00 |
| | P 8 | 0.79 | Good | 7.06 | 6.51 | 3.50 | 1.87 | 1.21 | 68.13 |
| | P 9 | 0.57 | Moderate | 7.17 | 4.48 | 3.04 | 2.95 | 14.68 | 198.38 |
| | P 10 | 0.86 | Good | 7.24 | 6.45 | 1.31 | 1.49 | 14.86 | 96.50 |
| | P 11 | 0.81 | Good | 7.91 | 7.94 | 3.80 | 2.39 | 18.95 | 49.73 |
| Winter 2017 | P 1 | 1.01 | Excellent | 7.02 | 9.72 | 2.60 | 0.97 | 4.34 | 551.40 |
| | P 2 | 0.52 | Moderate | 7.59 | 11.19 | 5.23 | 1.13 | 0.97 | 38.86 |
| | P 3 | 0.70 | Good | 7.66 | 11.67 | 1.32 | 0.68 | 3.12 | 9.61 |
| | P 4 | 0.92 | Excellent | 8.53 | 11.63 | 0.71 | 0.59 | 0.76 | 8.61 |
| | P 5 | 0.68 | Good | 8.65 | 11.68 | 0.17 | 0.68 | 5.65 | 9.39 |
| | P 6 | 0.36 | Poor | 8.00 | 10.33 | 1.99 | 0.66 | 6.19 | 14.24 |
| | P 7 | 0.81 | Good | 8.50 | 12.40 | 0.71 | 0.59 | 0.76 | 8.61 |
| | P 8 | 0.38 | Poor | 8.28 | 12.96 | 1.38 | 1.69 | 6.80 | 17.88 |
| | P 9 | 0.74 | Good | 8.04 | 13.58 | 1.25 | 2.48 | 6.89 | 20.35 |
| | P 10 | 0.89 | Excellent | 6.79 | 9.95 | 0.00 | 1.52 | 2.16 | 40.13 |
| | P 11 | 0.52 | Moderate | 7.70 | 11.99 | 0.00 | 2.61 | 4.90 | 8.65 |

*3.2. Results of the Partial Least-Squares Analysis*

By gathering the pressure values from Figure 3 for the drainage areas represented in Figure 1, and using the SWP and IPtI$_N$ values shown in Table 2, two datasets were created, one for each PLS-PM model, namely the short scale (M1) and long scale (M2) models (Figure 4). Both include the summer and winter sub-models. For model M1, the latent variable "Point Source Emissions" was discarded because effluent discharge points were missing in the 250 m buffer zones of the sampling sites. In both models, the yellow boxes represent measured variables (MV) and the blue circles represent latent variables (LV). The labelled links (arrows) between MVs and LVs are the weights attributed to each MV by the model and the measurement of the MV contribution to the associated LV. The labelled links between LVs are path coefficients, which represent direct causal effects between latent variables. The coefficients of determination ($R^2$) measure the overall robustness of these effects.

The differences among models are apparent in regard to season and scale. For example, in winter, the $R^2$ values of M2 are higher than in its counterparts in M1: in M2 the $R^2$ values of "Ecological integrity (W 17)" and "SWP (W 17)" are 0.657 and 0.923, respectively, while in M1 they are 0.544 and 0.635. In summer, this relationship reverses: the $R^2$ values of "Ecological Integrity (S 16)" and "SWP (S 16)" are 0.332 and 0.795 in M2 and 0.798 and 0.823 in M1. In both models, the $R^2$ values are higher for "SWP" than for "Ecological Integrity". The highest $R^2$ of all refers to "SWP (W 17)" in M2 (0.923), and the lowest refers to "Ecological Integrity (S 16)" in M1 (0.332).

To understand the contribution of a measured variable in the final value of "Ecological Integrity", it is necessary to consider its weight and trace total effects among latent variables. For example, in M2 (Figure 4), the direct effect of "Point Source Emissions" on "Ecological integrity (S 16)" is the path coefficient 0.107, which is unexpected. However, the total effect of "Point Source Emissions" on "Ecological Integrity (S 16)" comprises indirect effects [86], because "Point Source Emissions" constitutes "SWP (S 16)" (path coefficient = 0.330), which, in turn, constitutes "Ecological Integrity (S 16)" (path coefficient = −1.105). The product of these path coefficients is the indirect effect of "Point Source Emissions" on "Ecological integrity (S 16)" (0.330 × −1.105), which is −0.364. The total effect is the sum of the direct effect (0.107) and the indirect effect (−0.364), which is −0.257, as expected. The contribution of a measured point source emission is the product of the total effect and the weight. For example, the weight of "Effluents (hydric)" is 1.188. The contribution of "Effluents (hydric)" to "Ecological Integrity (S 16)" is −0.257 × 1.188 = −0.305, as expected. This analysis applies to

all measured variables and latent variables and greatly improves the interpretation of the models. Figure 5 summarizes the total contributions of the measured variables to the latent variables "SWP" and "Ecological Integrity". Blue and red represent winter and summer, respectively. The dotted and continuous lines represent short (M1) and long (M2) scale models.

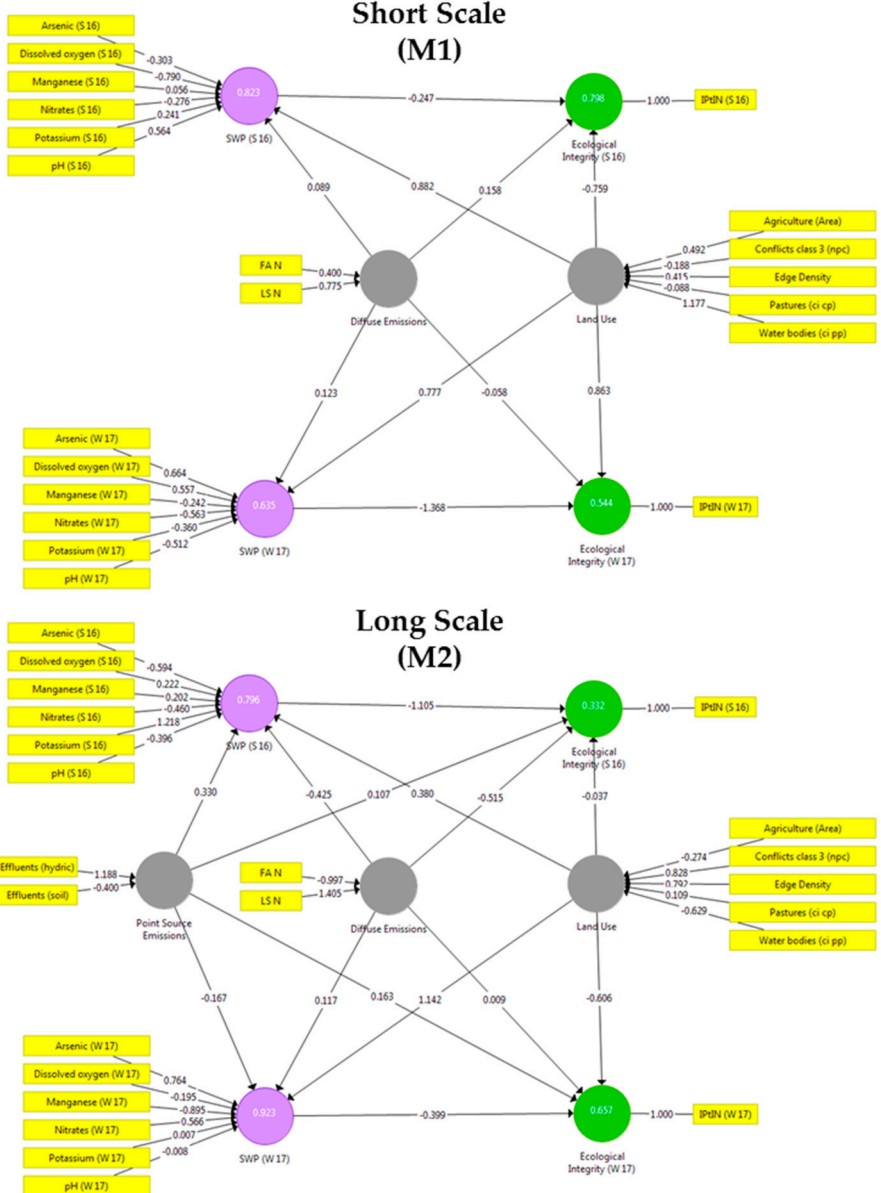

**Figure 4.** PLS-PM models. The short-scale model (M1, upper panel) involves the assessment of pressures within 250 m buffer zones around the sampling sites. The long-scale model (M2, lower panel) involves the assessment of pressures within the drainage areas located upstream of the sampling sites. Table 1 explains the symbols (e.g., FA N). Yellow boxes represent measured variables and blue circles, latent variables. Between measured variables and latent variables are represented weights, and between latent variables the path coefficients are established.

**Surface Water Parameters**

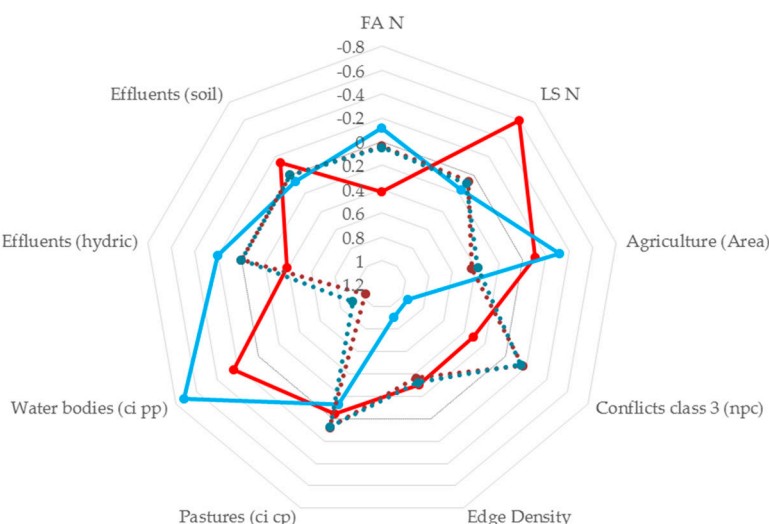

**Ecological Integrity**

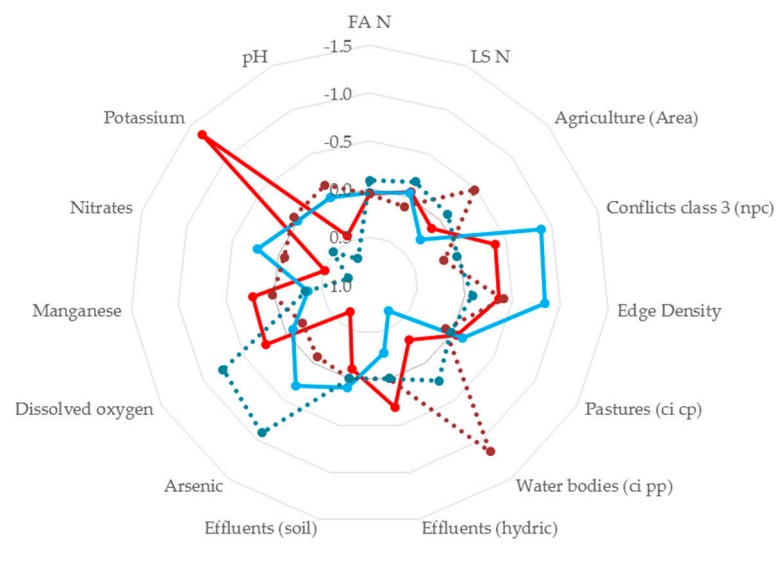

**Figure 5.** Contributions of the measured variables to the latent variables "SWP" and "Ecological integrity", in summer and winter and on short and long spatial scales.

In the short spatial scale, the effect of season on the relationship between anthropogenic pressures and surface water contamination is minimal, because the dotted lines practically overlap in the upper panel of Figure 5. For the opposite reason, in the long spatial scale, the effects of season are evident (compare continuous lines in the same panel). Major deviations occur for diffuse discharges of livestock nitrate ("LS N"), patches of severe land use conflict ("npc") and edge density, where the impact

is larger in winter. Some major deviations also occur for diffuse discharges of agro-forest nitrate ("FA N"), point source discharges of nitrate in the streams ("Effluents (hydric)") and water body patch connectivity "ci pp", where the impact is more evident in summer. The results also show differences across spatial scales. For example, there is a significant difference between the impact of "ci pp" in the short and long spatial scales.

The lower panel of Figure 5 illustrates the effects of anthropogenic pressures and surface water contamination on ecological integrity. Here, seasonal influences occur on the short and long spatial scales. For the short spatial scale, major deviations occur for "ci pp", which appears to impact ecological integrity more significantly in summer, and for "dissolved oxygen" and "arsenic", which seem to have greater effects in winter. For the long spatial scale, major deviations occur for "potassium" in summer and for patches of severe land use conflict ("npc") and "edge density" in winter.

## 4. Discussion

Taken together, the models seem to better explain the relationship between anthropogenic pressures and water quality than the influences of these latent variables on ecological integrity. Additionally, in winter, the long-scale model seems to behave better than the short-scale model, and in summer, the performance levels of the two models reverses.

Usually, path coefficients linking "Pressures" to "SWP" should be positive, because it is expected that increasing anthropogenic pressures will increase stream water contamination. In turn, path coefficients linking "Pressures" to "Ecological integrity" or "SWP" to "Ecological integrity" should be negative, because increasing human pressure and stream water contamination are expected to cause ecological integrity losses. In the present study, the signs of path coefficients are as expected with a few exceptions. In model M1, the exceptions are the links between "Diffuse Emissions" and "Ecological Integrity (S 16)" (0.158) and between "Land Use" and "Ecological Integrity (W 17)" (0.863). In model M2, the unexpected signs refer to links between "Diffuse Emissions" and "SWP (S 16)" (−0.425) and "Ecological Integrity (W 17)" (0.009), and between "Point Source Emissions" and "SWP (W 17)" (−0.167), "Ecological Integrity (W 17)" (0.163), and "Ecological Integrity (S 16)" (0.107). The unexpected signs of path coefficients may be a consequence of indirect effects between latent variables.

The surface water parameters that seem to have concurrent effects on ecological integrity in the M1 and M2 spatial scale models, while having opposite effects in the seasonal sub-models, are arsenic, manganese, and potassium.

Arsenic appears to be negative for ecological integrity during the winter period. This effect might be related to agricultural practice in the winter [91]. The use of phosphate fertilizers in agriculture is known to lead to the accumulation of arsenic in soils, water, and crops [92–94]. These agrochemicals are toxic, and various recent papers have emphasized this toxicity issue [95]. However, its use in the Sabor River Basin seems to have continued [68]. The arsenic concentrations are higher in summer than in winter. Therefore, the harmful effects of arsenic are related to factors other than the concentration that influence the bioaccumulation of this substance [96].

Potassium is an essential component of fertilizers used in agriculture, so its adverse effect on the latent variable "Ecological Integrity" during the summer period might be attributed to this activity [97]. In the Sabor River Basin, the production of olives and grapes occurs on a regular basis. The use of potassium is essential for the growth of these goods [98,99]. In the region, fertilization of oliveyards and vineyards occurs in the spring season when rainfall, runoff, and sub-surface flow may be abundant. The leach of potassium in runoff and shallow soil and saprolite waters can justify the high concentrations measured in the forthcoming season (summer).

The effect of manganese on "Ecological Integrity" is adverse during the summer period. The bedrock in the Sabor River Basin comprises manganese bearing rocks such as serpentinites [3]. In summer, the contribution of groundwater to streamflow is high because rainfall is lacking [100]. The discharge of groundwater naturally enriched in manganese can then accumulate in the macroinvertebrates reducing the $IPtI_N$ values. The response of $IPtI_N$ to changes in pH, dissolved oxygen,

and nitrate concentrations is not consistent across scales or seasons. The results for these parameters are, therefore, difficult to explain.

Some diffuse emissions revealed concordant effects on water quality at the seasonal level but opposite effects at the spatial scale level. Agricultural areas and water body connectivity were shown to contribute to water quality decline in the summer and winter periods, but only on the short spatial scale. On the long scale, the effect seemed to be neutral (agricultural areas) or even positive (water body connectivity), especially in winter. Adverse effects of agriculture patches have been reported elsewhere [38]. However, to the best of our knowledge, there have been no previous reports on negative effects of water body connectivity on water contamination. Eventually, disconnected water bodies are not able to dilute contaminants brought from upstream areas. This could be a reliable setting on a short spatial scale. Besides, water body connectivity increases the water quality on the long scale, a result that corroborates and completes this reasoning.

The landscape metrics "npc" (number of patches with a land use conflict of class 3) and "ci cp" (connectivity of pasture areas) were associated with adverse effects on water quality on the long spatial scale. In both cases, these areas are scattered within the Sabor River Basin and are characterized by small patch areas. For that reason, only the long scale analysis was able to trace them.

The metric "edge density" revealed a general negative effect on the ecological integrity in both seasons and spatial scales. The complexity of a landscape is intrinsically connected to the "edge density" metric. Higher values indicate a wider variety of land uses and irregular patches [101]. According to some authors, the increase in edge density is a consequence of human intervention in the landscape [31], which transforms natural habitats into agricultural and urban areas and influences the transport of nutrients and contaminants from soil to water [102]. In the Sabor River Basin, the "edge density" metric seems to be the prominent threat to ecological integrity, because it is independent of season and scale. This study revealed that the use of two PLS-PM models was enough to compare the seasonal and spatial scale effects of anthropogenic pressures on water quality and ecological integrity. Frequently, the assessment of water contamination and ecological integrity involving landscape metrics uses a very large number of these indices [30,38]. The present study was based on five landscape metrics, because multicollinearity is a crucial issue in PLS-PM and many landscape metrics are collinear. In spite of using just a few landscape metrics, this study had the advantage of combining these important variables with other parameters relevant for water quality, such as point source and diffuse emissions. This combination is rarely seen in other studies focused on relationships between anthropogenic pressures, water quality, and ecological integrity.

One weak point of this study is the small sample size (just 11 entries). We note that PLS-SEM can handle efficiently small samples with a large number of variables, which is preferable over other structural equation (SEM) models (e.g., CB-SEM) [59]. A larger number of samples would increase parameter estimates (e.g., $R^2$), which are already satisfactory or even good in most cases. Further studies on this or other river basins should, therefore, be based on a larger number of sampling sites.

### 4.1. Scope of Study and Environmental Management Guidelines

As the measurements of $IPtI_N$ revealed, the ecological status of Sabor River Basin is under acceptable conditions. Compared to other river basins under poor ecological status, such as the Ave River Basin, it is eminent to mitigate pollution effects looking forward to achieving a better ecological status [103]. Anyhow, it is important to preserve such conditions in Sabor River Basin at points where the ecological status is under "good" status and improve some sampling sites where an ecological status is below the "good" classification. For such reasons, the present work fits within the scope of 2030 Agenda Sustainable Development Goals, announced by the United Nations, by adopting goal 15, called "Life On Land". From an overall perspective, this goal attracts worldwide attention to preserving natural resources and promote natural biodiversity in all ecosystems. In addition, it is given special attention to terrestrial ecosystems, including inland freshwater, for the purpose not only of restoration but also of preservation.

With regard to river basin management, this study contributes to such cause, as the pressures that degrade ecological integrity are identified, as well as the most vulnerable season. It was also identified the scale at which pressures should be addressed, advising that, for other studies of the same scope, the scales where there is a negative effect of the pressures, should be applied. In this context, the results obtained from Figure 5 were compiled into Table 3, where are shown the scales and seasons to be given particular focus by decision-makers are identified.

**Table 3.** Pressure effects in ecological integrity, is marked with an x the season and scale were it was identified a negative effect.

| | Long Scale | | Short Scale | |
|---|---|---|---|---|
| | Summer 2016 | Winter 2017 | Summer 2016 | Winter 2017 |
| FA N | | | | X |
| LS N | X | X | | X |
| Effluents (hydric) | X | | | |
| Effluents (soil) | | X | | |
| Agriculture (Area) | | | X | X |
| Conflicts class 3 (npc) | X | X | | |
| Edge Density | X | X | X | X |
| Pastures (ci cp) | X | X | | |
| Water bodies (ci pp) | | | X | X |

The identified pressures mainly have a negative effect throughout the seasons. Nevertheless, it should be noted that nitrogen discharges from agricultural areas should be given particular attention during winter. This is probably due to fertilization periods, analyzed on a short scale, the effect is negative. Furthermore, this indicates that when quality measurements were taken nearby, the Directorate-General for Agriculture and Rural Development (DGADR) [104], in cooperation with the Portuguese Environmental Agency (APA) [73], should supervise the fertilization period to minimize the effects of excessive fertilizer application [105].

Nitrogen discharges from livestock farming should be monitored during winter and summer to ensure that contamination does not spread through surface runoff or groundwater contamination. Regarding this environmental problem, several preventive measures must be taken, aiming at sustainable livestock production [106]. When livestock production farms are installed, the site selection is a crucial aspect, in order to allocate the farms distant from wells, in areas with low soil permeability and a low slope to control the runoffs of agricultural fertilizers [107]. The site selection is a crucial aspect, in order to allocate the farms distant from wells, in areas with low soil permeability and a low slope to control the runoffs of agricultural fertilizers [108].

The Sabor River Basin contains mostly urban effluent discharge points. In this study, the analysis was done only on a large scale, as there were no discharge points on the short scale. The effect of effluent discharges into surface waters has been found to degrade quality during the summer period, whereas discharges to the soil only have a negative impact during the winter period. This information is useful for monitoring and management purposes. By knowing which is the season that effluent discharges have a negative impact on water quality, environmental entities should enforce inspections, looking forward to overseeing if discharge values are under legal thresholds, even if the treatment station is operating properly.

The effects of land use metrics in ecological integrity shown clearly dependence on scale and season. As other authors have shown is seen that when a scale is accessed, a positive effect can be detected while, on another scale, a negative effect can be detected [32,34,109]. Agriculture (areas) and Water bodies (ci pp) only have a negative impact on a short scale, conflict class 3 (npc) and Pastures (ci cp) have a negative impact on a long scale, and only Edge Density revealed a negative impact in all scales and seasons. Such results show the effort of geospatial data effects, which should be applied at various scales. In terms of river basin management, the landscape variables revealed a stronger effect on ecological integrity when compared to diffuse and point source emissions. Conflict class 3 (npc)

revealed the strongest effect in the long-scale (−0.378 in S 16 and −0.880 in W 17) and Water Bodies (ci pp) in the short-scale (−1.150 in S 16 and −0.236 in W 17). In the (PGRH) Portuguese Management Plan of Hydrographic Regions are identified many threats to the ecological integrity of Sabor River Basin, namely the diffuse discharges of livestock and urban effluents. By default, the plan does not include intervention measures based on land use configuration. As shown in this study, land use configuration is a key aspect that overlaps the effect of effluent discharges. In this context, it is crucial that such studies where diffuse and point discharges are analyzed should add the metric effects of land use to access a wide range of pressures and phenomena that affect the ecological integrity. Even more, it is understood that in the PGRH should be implemented territorial planning as a measure to improve water quality at a national level.

Land use changes in the Sabor River Basin would be a step of ecological quality assurance. On a large scale, land use transitions can be expensive and take long periods to implement [110], but strategic reordering can be a key process. Even so, the places where change can be less expensive are forest and agricultural areas. In the results, it was found that areas with land use conflict class 3, which are agricultural areas, have a strong impact on water quality. One of the measures is to propose the reallocation to areas where the land is suitable for such practices. As it is detected in Figure 3E, areas with land use conflict are small patches strategically located and easily replaced by forestry land uses.

In the present study, the measured concentration of nitrates fulfilled the Portuguese legislation threshold (minor to 25 mg/L) and subsequently the Nitrates Directive (2006/118/EC) (ND) requirements for drinking purposes (minor to 50 mg/L). However, stricter benchmarks should be addressed, since a concentration minor to 3 mg/L indicates contamination [111], and the United States Environmental Protection Agency maximum concentration level for nitrate is 10 mg/L in order to protect against blue-baby syndrome [112]. As it was identified in this study, the nitrogen sources are affecting Sabor River Basin water quality. By identifying the effects of each pressure, the applied methodology revealed itself as an important tool for the protection of aquatic systems, even when contaminant concentration fulfil legal limits. Notwithstanding that the Sabor River Basin is under an acceptable ecological status, the strategies to reduce the diffuse emissions from agriculture and livestock (presented in the ND) and complementary strategies (presented in the WFD) [113] should be taken into action, in order to preserve water quality.

## 4.2. Study Limitations and Future Recommendations

There are essential aspects for unerring river basin monitoring, such as the number and spatial distribution of sampling sites, the number of measurements made at each point and analysis method [114], which may consist on a statistical or physical approach [115]. Such elaborated methodologies are craved by researchers and stakeholders but, nevertheless, it is crucial to have proper funding [116] for the elaboration of extensive and detailed sampling campaigns, or even to achieve detailed datasets, by the application of in situ automated samplers [117]. The present study design is quite innovative since it compares simultaneously the scale and seasonal effects in water quality through an explicative statistical approach. Still, it is alerted that some aspects of the present work could not be optimal due to logistic issues, namely the number of sampling sites and sampling frequency. Besides PLS-PM requires a smaller sample size than other statistical methods, such as traditional SEM [118], is preferable to use a higher number of samples. This aspect could be improved by increasing the number of sampling sites in Sabor River Basin, covering a greater part of the river basin. It is not expected that the cause-effect relationships would change, since the sub-catchment areas are predominantly rural, but it would improve statistical significance [119]. The present study pictured the interplay among landscape metrics, pollution sources with water quality parameters, through water quality samples that were measured simultaneously with $IPtI_N$ in each sampling site. For such reason, our results can exhibit linkages between variables that were contained in the dataset, but this cannot be seen as undisputed interactions for Sabor River Basin. To achieve such evidences, it would be necessary

the sampling frequency by achieving a more detailed timestep, by using seasonal, or even monthly averages, that would give a more detailed picture of Sabor River Basin environmental interactions. For such reasons, some monitoring technical aspects are recommended, in order to enhance the virtue of future studies. By increasing the number of sampling sites it is possible to achieve high reliability when sampling by using at least 20 sampling sites strategically distributed [120]. To ensure reliability is crucial to have a detailed time step to monitor contaminant concentrations. The adequate sampling frequency is discussed by many authors and is also dependent on the type of contaminant [121], but sampling frequencies such as weekly to bi-monthly can be suitable for monitoring [122], supported by other author results [123] is advised that, for contaminant concentrations, the sampling frequency should be weekly. Since benthic macroinvertebrates communities take time to assemblage, comparing the changes of lifeforms across a short time step might be meaningless [124]. For such reasons is recommended that the IPtI$_N$ sampling should be done monthly, since it was already revealed that the presence of such lifeforms can change in periods of two months [125].

## 5. Conclusions

This study, which was based on partial least squares-path modeling (PLS-PM) and conducted in the Sabor River Basin (Northeast Portugal) revealed significant seasonal and spatial scale influences of anthropogenic pressures on stream water quality and ecological integrity (measured by a macroinvertebrate biodiversity index). The most influential pressure was the metric "edge density" because it was associated with ecological integrity declines on both short (within and 250 m around target sites) and long spatial scales (within the drainage area upstream of the target site), as well as in winter and summer. Other pressures were also important but influenced water contamination or ecological integrity differently on the short vs. long scale or in winter vs. summer. For example, agricultural area patches and water body connectivity were shown to be adversely related to the short spatial scale but positively related to the long spatial scale. Land use conflicts (deviations of actual from natural uses) as well as pasture area connectivity were shown to adversely affect the ecological integrity solely on the long scale. Finally, surface water parameters, such as arsenic, potassium, and manganese, were shown to adversely affect the ecological integrity on both the short and long spatial scales, but the effects were different in summer vs. winter. The explanations for the seasonal differences were diverse for the three parameters, for example, seasonality in the application of farmland fertilizers, geology, and the share of surface water and groundwater in stream flow. Overall, the study exposed a complex interplay among anthropogenic pressures, stream water contamination, and ecological integrity, which was uncovered using PLS-PM. This sophisticated statistical model proved to be efficient for the assessment of these intricate relationships, but could also be used for the complex management of multiple land use watersheds such as the Sabor River Basin. As it was identified in another study, Sabor River Basin is under acceptable ecological conditions (according to data collected in 2012) [64]. However, as it was revealed in this study (with data from 2016 and 2017), it is crucial to monitor point source emissions and to decrease nutrient loads from diffuse emissions and reforest agricultural areas that are under land use conflicts in order preserve such ecological status, or even to improve it.

Our results, therefore, present a challenge to municipal and regional water planners, as well to water management authorities, which are responsible for the safeguarding of water quality.

**Author Contributions:** conceptualization, A.C.P.F.; methodology, A.C.P.F. and F.A.L.P.; software, A.C.P.F.; validation, F.A.L.P.; formal analysis, L.F.S.F.; investigation, A.C.P.F. and D.P.S.T.; resources, L.F.S.F. and R.M.V.C.; data curation, A.C.P.F., F.A.L.P., L.F.S.F. and R.M.V.C.; writing—original draft preparation, A.C.P.F.; writing—review and editing, F.A.L.P.; visualization, A.C.P.F. and D.P.S.T.; supervision, F.A.L.P. and L.F.S.F.; project administration, R.M.V.C.; funding acquisition, R.M.V.C.

**Funding:** This research was funded by the INTERACT project—"Integrated Research in Environment, Agro-Chain and Technology", no. NORTE-01-0145-FEDER-000017, in its line of research entitled BEST, co-financed by the European Regional Development Fund (ERDF) through NORTE 2020 (North Regional Operational Program 2014/2020). For authors integrated with the CITAB Research Centre, it was further financed by the FEDER/COMPETE/POCI—Operational Competitiveness and Internationalization Programme, under Project POCI-01-0145-FEDER-006958, and by the National Funds of FCT—Portuguese Foundation for Science and Technology, under the project UID/AGR/04033/2019. For the author integrated in the CQVR, the research was additionally supported by the National Funds of FCT—Portuguese Foundation for Science and Technology, under the project UID/QUI/00616/2019.

**Conflicts of Interest:** The authors declare no conflict of interest. The funders had no role in the design of the study; in the collection, analyses, or interpretation of data; in the writing of the manuscript or in the decision to publish the results.

## Appendix A

The calculation of land use conflicts was based on a simple methodology, where the land capability (natural use) was compared with the actual land use [39]. In this method, the land capability is calculated by the ruggedness number (RN), which is the product of the drainage density and the terrain slope (as a percentage) [126]. The RN was classified into four classes through natural breaks. Each class is given a score from 1 to 4, where 1 represents the areas with the lowest RN value, which are suitable for agriculture or any other land use. For scores 2, 3, and 4 the land use is suitable for pasture (livestock production), a mosaic of pasture and forest, or just forest, respectively. The land use map is reclassified with the same scoring procedure. The land use conflict class is set as the difference between the land capability score and the actual land use score, which varies from 0 to 3. For the highest conflict class (3), the recommended use is forestry to protect the soil. The other conflict classes are recommended for other less protective uses. When the difference between capability and land use is 0 or negative, there is no conflict between the natural and actual land uses. Table A1 shows the land use class conflict, based on land capability and land use, in classes ranging from 0 to 3. The evaluation of land use conflicts has already been used in a variety of studies [9,40,43,44,46,48], in terms of soil and hydric resources conservation.

**Table A1.** Classification of land use conflict class according to land use and capability.

| Land Capability ╲ Land Use | Agriculture | Pasture | Pasture/Forestry | Forestry |
|---|---|---|---|---|
| Agriculture | 0 | 0 | 0 | 0 |
| Pasture | 1 | 0 | 0 | 0 |
| Pasture/Forestry | 2 | 1 | 0 | 0 |
| Forestry | 3 | 2 | 1 | 0 |

## Appendix B

Water quality samples were collected from the 11 sites and analyzed during the summer of 2016 and the winter of 2017, according to the Portuguese water masses classification handbook [54] (title in English: Criteria for the Classification of the State of the Surface Water Masses—Rivers and Reservoirs). The guidelines presented in the document were created according to the monitoring program of the Water Framework Directive (Directive no. 2000/60/CE) transposed to the Portuguese legislation (in law no. 58/2005 and Decree-law no. 77/2006). In surface water, physico-chemical parameters, dissolved oxygen, temperature, pH, total suspended solids, and conductivity were measured in the field using a multi-parameter probe. For the other parameters (presented in Table 1), water samples were collected and stored in the dark and under refrigeration prior to measurement in the laboratory using ion chromatography (Dionex Equipment).

**Appendix C**

The North Invertebrate Portuguese Index (IPtI$_N$) is commonly used to evaluate the ecological status of Portuguese waters in the north zone, while an identical index is used for the south zone (IPtIs), due to the biological and climate differences between the two zones of the country. The calculation of this index is based on the presence of benthic invertebrates, due to their sensitivity to a vast spectrum of pollution sources [54].

To perform the measurement of this indicator, organism samples were collected from 11 strategic surface water locations (Figure 1C). For each sample, the organisms were classified and then counted. Equation (1) was used to calculate IPtI$_N$ at each site. The equation used is quite complex, since it uses diverse parameters, such as the number of taxonomic groups present in the sample (N° Taxa), the number of families that belong to the *Ephemeroptera*, *Plecoptera*, and *Trichoptera* orders (EPT), the Pioleu Index or Evenness [127,128], the biological monitoring working party index divided by the number of families included in this index (IASPT) [129], and the sum of individuals belonging to the *Heptageniidae*, *Ephemeridae*, *Brachycentridae*, *Goeridae*, *Odontoceridae*, *Limnephilidae*, *Polycentropodidae*, *Athericidae*, *Dixidae*, *Dolichopodidae*, *Empididae*, and *Stratiomyidae* families (Sel.ETD):

$$\text{IPtI}_N = \text{N° Taxa} \times 0.25 + \text{EPT} \times 0.15 + \text{Evenness} \times 0{,}1 + (\text{IASPT} - 2) \times 0.3 + \text{Log (Sel.ETD} + 1) \times 0.2 \qquad \text{(A1)}$$

$$\text{IPtI}_S = \text{N° Taxa} \times 0.4 + \text{EPT} \times 0.2 + (\text{IASPT} - 2) \times 0.2 + \text{Log (Sel.EPTCD} + 1) \times 0.2 \qquad \text{(A2)}$$

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
