# Peer review of "Seasonal and Scale Effects of Anthropogenic Pressures on Water Quality and Ecological Integrity: A Study in the Sabor River Basin (NE Portugal) Using Partial Least Squares-Path Modeling"

_water, doi:10.3390/w11091941_

Round 1

Reviewer 1 Report

It is an interesting study, but poorly and insufficiently explained. Some problems are given below

1. Problem identification is poor. It mentioned that “The ecological status is good in 92 approximately 80% of all Sabor streams and streamlets (Line 95-95)”, so what motivated this study is not clear.

2. In section 2.2, it is written “The second step continued with estimation of an average value for each pressure, within the buffers and the drainage areas, while twenty-two surface water parameters and a biological index (North Invertebrate Portuguese Index - IPtIN) were evaluated at the sampling site in the summer of 2016 and winter of 2017.”

Please name those 22 surface water parameters.

3. In Line 194, it is written “Table 2 depicts the scores of water quality parameters and of IPtIN index”. However, what shown in the table appears to be concentrations of the six parameters. Does the “score” simply mean the concentration? Besides, parameters such as dissolved oxygen vary in space and time, are they sort of averaged values? Furthermore, Nitrate in Table 2 is NO3-N or nitrate ion?

4. How the environmental land use conflicts were defined?

5. Give the definition for edge density

6. Overall, this work looks more like an exercise for a statistical analysis than a water quality and ecological study. There is little useful information for better river management

Author Response

It was attached to this document the MDPI English editing certificate

Problem identification is poor. It mentioned that “The ecological status is good in 92 approximately 80% of all Sabor streams and streamlets (Line 95-95)”, so what motivated this study is not clear.

Many thanks for this comment. It is rather pertinent and constructive. Environmental studies should not only aim for remediation/mitigation purposes. Such methodologies should be applied not only to identify the main pressures that degrade water quality in widely contaminated sites, such as the Ave River Basin, but also to preserve the quality of systems that may be degraded. The purpose of environmental studies is not only to intervene but also to prevent the loss of water quality. It was added more information about the study area (lines 132-139) to show such motivation in the manuscript.

In section 2.2, it is written “The second step continued with estimation of an average value for each pressure, within the buffers and the drainage areas, while twenty-two surface water parameters and a biological index (North Invertebrate Portuguese Index - IPtIN) were evaluated at the sampling site in the summer of 2016 and winter of 2017. Please name those 22 surface water parameters."

Actually, there are 24 parameters, in the text it was mentioned that they were 22 parameters, but it has already been corrected to 24 (lines 152-153). It was also made reference to table 1 where the 24 parameters listed:

1- pH; 2- temperature; 3- conductivity; 4- dissolved oxygen; 5- nitrites; 6- nitrates; 7- sulfates; 8- phosphates; 9- total suspended solids; 10- calcium; 11- iron; 12- magnesium; 13- potassium; 14- sodium; 15- total aluminum; 16- arsenic; 17- cadmium; 18- lead; 19- cobalt; 20- copper; 21- manganese; 22- zinc; 23- nickel and  24- chromium.

In Line 194, it is written “Table 2 depicts the scores of water quality parameters and of IPtIN index”. However, what shown in the table appears to be concentrations of the six parameters. Does the “score” simply mean the concentration? Besides, parameters such as dissolved oxygen vary in space and time, are they sort of averaged values? Furthermore, Nitrate in Table 2 is NO3-N or nitrate ion?

Corrected, it was intended to write concentration of SWP and IPTIN score, (line 242). Nitrates are in the form of NO3- ion, in table 2 it was added the chemical symbols (line 256).

Both SWP and IPTIN were measured once in each site for each season. The table only depicts measured values, during the sampling campaign. It would be better to extend the sampling campaign, measuring the parameters more than one day during each season but unfortunately, that would be costly exceeding the budged allocation to this task in the funding project.

How the environmental land use conflicts were defined?

It was written more information about land  use conflicts in introduction and also in appendix A (lines 77-79) and (lines 519-535)

Land use conflics are the difference between the land capability and actual land use. This methodology is applied according to the heavy demands that agricultural areas place in soil and forestry areas conserve it. In terms of land use and land capability is given a score from 1 to 4, agricultural areas (score =1), pastures for livestock production (score =2),  transition areas (forestry or pastures) (score =3) and forest score (score=4). By subtracting the land capability score to the land use score is given the land use conflict. Example, if an area has only capability for forest (score=4) and is occupied by an agriculture field (score =1) the land use conflict class is equal to 3. But if an area has suitability for any land use, each means that is suitable for agriculture (score=1) and is occupied by forestry (score =4) the land use conflict class is not -3 but 0, since in such methodology the classification goes from 0 to 3. This information is resumed in table A.1. line 535.

Give the definition for edge density

In lines 201-202 it was added  information about the calculation procedure of edge density. Which is the sum of all patches length and then divided by the area.

Overall, this work looks more like an exercise for a statistical analysis than a water quality and ecological study. There is little useful information for better river management.

In this study it was intended to demonstrate the cause-effect relationships of point source and diffuse  source contamination, landscape metrics on SWP and ecological integrity, and also how these cause-effect interactions can change through season and scale. With such variables, it was intended to use multivariate statistics to exhibit all the interactions. By using  PLS-PM it was demonstrated all the interactions changes through season and scale demonstrating in which season should be prioritized proper monitoring, and which is the scale that reveals the most harmful effect.

At the end of discussion (lines 397-478) it was added a new topic   “4.1 Scope of Study and Environmental Management Guidelines“. In this topic it was included information for proper river management. Which is the most relevant information for decision-makers. The proposals were made according to the effect of the analyzed pressures on ecological integrity.

Reviewer 2 Report

Thank you for submitting your manuscript to the Water journal. Generally, the manuscript fits into the scope of the journal and the structure respects Scientific Best Practice. However, there are some comments that require revision.

In the title you should not use abbreviations (PLS-PM), as the general reader might not understand the title. Further, you should check if you use abbreviations in the abstract.

In the introduction, you need to connect the state of the art to your manuscript goals. Please follow the literature review by a clear and concise state of the art analysis. This should clearly show the knowledge gaps identified and link them to your manuscript goals. Please reason both the novelty and the relevance of your manuscript goals. In the end of the introduction section you should clearly explain the gaps from the literature and conclude the need to close the gaps. In this frame you need to explain the scope of tthe work presented.

In the methodology section must be indicated the source of all figures. In the text is used the term ecological integrity without explanation. The definition must be added. Also more information on the model must be added.

Futher, in the methodology section, the water sample detection and analysis: have been applied analysis approaches according to DIN ISO ? If yes, which ? If not, why not ? What ist he uncertainty of the analysis procedures ? This information might be also included into appendix B. How was done the quality assurance, if the analysis was not done according to DIN ISO ? How do you ensure statistical reliability?

In the results section tables shall not be split on two pages (see table 2). The coloured parts must be explained in the legend of the table.

In the discussion section should be made reference to the EU nitrate strategy requirements, the EU Nitrate Directive and the implications for the current case study.

In the conclusions, in addition to summarising the actions taken and results, please strengthen the explanation of their significance. It is recommended to use quantitative reasoning comparing with appropriate benchmarks, especially those stemming from previous work.

Author Response

It was attached to this document the MDPI english editing certificate

1- Thank you for submitting your manuscript to the Water journal. Generally, the manuscript fits into the scope of the journal and the structure respects Scientific Best Practice. However, there are some comments that require revision.

We very much thank the general good appreciation made by the reviewer about our study. All the required changes were made in the manuscript.

2-In the title you should not use abbreviations (PLS-PM), as the general reader might not understand the title. Further, you should check if you use abbreviations in the abstract. 

We changed the title according to the reviewer suggestion. The new title is:

“Seasonal and scale effects of anthropogenic pressures on water quality and ecological integrity: A study in the Sabor River Basin (NE Portugal) using Partial Least Squares - Path Modeling”

It was also rephrased the abstract nomenclature, lines(19-20)

3- In the introduction, you need to connect the state of the art to your manuscript goals. Please follow the literature review by a clear and concise state of the art analysis. This should clearly show the knowledge gaps identified and link them to your manuscript goals. Please reason both the novelty and the relevance of your manuscript goals. In the end of the introduction section you should clearly explain the gaps from the literature and conclude the need to close the gaps. In this frame you need to explain the scope of the work presented.

The application of structural equation models is a method that is taking its first steps in the environmental sciences, as few authors have adopted this statistical method. In terms of scale, it is known that variations of the scale accessed, watershed, buffer, riparian have can affect the results. In the present study, it was intended to exhibit how the accessed scale can affect environmental interactions and also the effect of season in such interactions. In lines 65-74 it was added information to reinforce the problem of the scale effect,  and the relevance of the manuscript goals was enforced in the end of the introduction (lines 113-118)

4- In the methodology section must be indicated the source of all figures. In the text is used the term ecological integrity without explanation. The definition must be added. Also more information on the model must be added.

All the figures were created by the authors for this study. In figure 2 it was cited the handbook of iptin classification, such reference was removed since it is not necessary and is already refered in methodology and introduction.  The capitation of the figure2 was edited , adding more information  about the models. In the introduction it was given a brief definition of “Ecological integrity” (lines 85-92), it was not defined in the methodology section since the terms already used in the introduction.

5- Futher, in the methodology section, the water sample detection and analysis: have been applied analysis approaches according to DIN ISO ? If yes, which ? If not, why not ? What ist he uncertainty of the analysis procedures ? This information might be also included into appendix B. How was done the quality assurance, if the analysis was not done according to DIN ISO ? How do you ensure statistical reliability?

The sampling methodology was applied according to the Portuguese water masses classification handbook(title in English: Criteria for the Classification of the State of the Surface Water Masses—Rivers and Reservoirs). Samples were gathered in the same day. It would be better to increase the measurements for more than a single day, in order to produce daily or monthly averages. Unfortunately, that would be costly exceeding the budged allocation to this task in the funding project, and would not allow us to go into more detailed procedures.

6- In the results section tables shall not be split on two pages (see table 2). The coloured parts must be explained in the legend of the table.

Table 2 was replaced in order to keep it in a single page, in the captionof figure 2 it was added information about coloured parts.

7- In the discussion section should be made reference to the EU nitrate strategy requirements, the EU Nitrate Directive and the implications for the current case study.

It was added  new topic in discussion “4.1 Scope of Study and Environmental Management Guidelines“,referring strategies to improve Sabor river ecological quality does, according to the impacts of the analised pressures, lines(397-479)

8- In the conclusions, in addition to summarising the actions taken and results, please strengthen the explanation of their significance. It is recommended to use quantitative reasoning comparing with appropriate benchmarks, especially those stemming from previous work.

It was added more information in  conclusions, lines (501-506).

Reviewer 3 Report

  Application of the SEM (particularly PLS-PM) method in the assessment of anthropogenic pressures on water quality and ecological integrity gets more and more popular in scientific papers due to a number of advantages comparing  to e.g. other ordination methods. Although the findings are generally interesting, the results are somewhat expected, given previous literature on the subject, [included almost 50% of citations of works written by the authors themselves (!)], and many more that can be found by simple literature search. Therefore, another case-study report, while supporting previous evidence, adds quite little to the current knowledge on interaction between anthropogenic pressures and water quality and ecological integrity.

What would be of greater interest, in the context of applying PLS-PM model, is how the applied method can be used to eliminate harmful effects on water ecosystems or at least to show the advantage of the PLS-PM to practice.

My main criticism  refers, however, to sampling regime limited to two seasons S and W for which, the authors used results of 1 sample each. This is ad-hoc sampling, not methodologically sound. Taking altogether 2 samples only per sampling point, the authors fell in a common trap that  the results of water samples show a high degree of randomness. From methodological point of view cannot be acceptable.

This is the reason, that the study is not  accurately performed, without going deeper into the background and  the study seems to be yet another investigation supportive of rather known association between pollutant (pressure)  and water quality/ecological integrity (response) based on well known mechanisms, with the one sample comparison effect being among the major ones.

No scale effect is justified in this case.

In future, please go deeper into the details more detailed data and detailed analysis in subgroups of samples, it is critically required.

In spite of declining the ms from publication in Water journal, I have made some remarks which may help authors to publish the submitted ms in the future.

The structure of the ms is Ok, however the text needs extensive editing, also in terms of language (for example: not “phenomenons” (l.16) but phenomena).

My main remarks are related to:

Abstract(l.15-29) : is not informative enough. Aim of the ms needs to be clarified. Methodological aspects need to be more clear. Results should be reworked and “take-home message” indicated.

Introduction:

l. 39 Do really organic loads are contained in urban effluents? Please  be precised.

l. 48. What kind of landscape metrics do the authors mean? Please clarify

l. 77. The meaning of “Ecological integrity” needs clarification in the Introduction section. What is the reason to use PLS models to test it?

l. 84 Hydrographic Region: please give the reference

l. 87 and 89 How it is possible to consider the land use of the Sabor River Basin as rural while the share of land used by agriculture is 35%?

l. 91 average temperature : for which period: month, year, season?

Fig 1  Elevation color gradient is misleading: mountains and highlands are in red, orange and yellow, while lowlands are in green. Watershed divide ( not in legend) should be in red. What are the dashlines in fig 1c.? Moreover  figure caption 1c is not velar: what does it mean: “from this point to the drainage areas”?

l. 98: Workflow:  please change into a standard section heading “Materials and methods”

l. 103-113: Please provide relevant methodology of IPtIs. Macrozoobenthos sampling (2 samples) form P1 are rather ad-hoc and cannot be methodologically justified in this kind of study (to small N).

l. 126 Fig 2  What is the intention to provide the figure? What is its mission?

l. 137. Unecessary figure  – delete it please.

l. 185: After Terencio please add: et al.

Results

l.195 Please refer to EU standards for drinking water, as well.

l.96 “more strict” than what?

Fig 3.: legend need to be more specified:  What does it mean discharge (kg/km2). Do the author mean N load, maybe? What are the land use conflict classes

l. 210-226 No reference to fig 4 (probably). The description of fig 4 should include the interpretation of the drawing.  In its current form, it unnecessarily repeats the content placed on it.

Discussion needs more data-supported conclutions.

Author Response

It was attached to this document the MDPI English editing certificate

1- Application of the SEM (particularly PLS-PM) method in the assessment of anthropogenic pressures on water quality and ecological integrity gets more and more popular in scientific papers due to a number of advantages comparing  to e.g. other ordination methods. Although the findings are generally interesting, the results are somewhat expected, given previous literature on the subject, [included almost 50% of citations of works written by the authors themselves (!)], and many more that can be found by simple literature search. Therefore, another case-study report, while supporting previous evidence, adds quite little to the current knowledge on interaction between anthropogenic pressures and water quality and ecological integrity.

Many thanks for this comment. It is rather pertinent and constructive. In socie-economic sciences, there are numerous applications of SEM models. The application of structural equation models in an environmental scope is taking the first steps since few authors used this method to access environmental interactions. As in other studies mentioned, resorting to PLS-PM is possible to establish the cause-effect relationships. We fully agree that in the eyes of readers, methodologies are generally close. But when analysed with  detail, the studies different. In one of the first applications, Fernandes 2018 used the models to compare their application in rural and anthropogenic basins to test the performance of SEM-PLS models. In the present study, the models were used to verify the variation of the cause-effect relationships in different stations and measurement scales. And as the results have shown not all pressures decrease ecological integrity, it can be tracked seasonal variations and also is shown that the accessed scale can affect the results.

2- What would be of greater interest, in the context of applying PLS-PM model, is how the applied method can be used to eliminate harmful effects on water ecosystems or at least to show the advantage of the PLS-PM to practice.

Such information is important for decision makers. Due to this recommendation, it was added table 3 (line 478). It resumes the pressures that harm ecological integrity according to the season and accessed scale, also was added a new topic in discussion, section 4.1.

3- My main criticism  refers, however, to sampling regime limited to two seasons S and W for which, the authors used results of 1 sample each. This is ad-hoc sampling, not methodologically sound. Taking altogether 2 samples only per sampling point, the authors fell in a common trap that  the results of water samples show a high degree of randomness. From methodological point of view cannot be acceptable.

We fully agree that increasing the number of sampling sites and sampling frequency would strength this study. But unfortunately that would be costly exceeding the budged allocation to this task in the funding project. Anyhow it was accessed a Portuguese database SNIRH ( available at https://snirh.apambiente.pt/) were there are measurements of more than 300 parameters across 2000 stations spread across Portugal,  22 of these stations take place in Sabor River Basin. The data of the same SWP parameters were gathered and compared and the resulting values concordant to the SNIRH analysis, since are inside typical values of Sabor River Basin. Only for potassium, it was not found enough data, since the measurements by SNIRH only took place in two sites for a short period of time. Please see Table1 presented in this topic, not in the manuscript.

            Table1.Maximum, minimum and average values of SWP.

Arsenic

Manganese

pH

Dissolved oxygen

Nitrates

Potassium

µg (As)/l

µg (Mn)/l

mg (O2)/l

mg (NO3)/l

mg (K)/l

Measured
for this
study

Maximum

18.95

551.40

8.78

13.58

15.05

4.05

Minimum

0.03

8.61

6.79

4.03

0.00

0.51

Average

5.39

74.13

7.81

9.33

2.81

1.47

SNIRH

Maximum

24.00

900.00

8.96

14.80

39.10

x

Minimum

1.00

3.20

3.40

2.85

0.01

x

Average

5.64

44.08

7.09

9.12

2.65

x

4- This is the reason, that the study is not  accurately performed, without going deeper into the background and  the study seems to be yet another investigation supportive of rather known association between pollutant (pressure)  and water quality/ecological integrity (response) based on well known mechanisms, with the one sample comparison effect being among the major ones.

Besides is expected that pressures decrease water quality this study revealed that this interactions change across season and scale. In a study of Fernandes et all 2018 it was revealed that the interactions are quite different across river bains. In this study is intended to reveal that such interactions can also change along the accessed scale and also season. Such works that reveal how this interactions change in a quantitative scope are usefull to identify locations, seasons and spatial scales, that require special attention by researchers and decision makers.

5- No scale effect is justified in this case.

The scale effect is accessed in this study since two models were elaborated, one containing the pressure data at a distance of 250 meters upstream and the other model containing the whole drainage area of each sampling site. Of course, the use of only two distances can be seen as incomplete, the incorporation of other scales such as would be relevant, but it would bring too much information to this study, and would be difficult for readers to interpret. It was adopted to use only two scales, because in these were tracked changes in the cause-effect relationship, as well as the effect of the seasons. But already in other studies, where scaling effects are acessed, only two distances are studied. The lines (65-74) have been added in the introduction to focus on this problem.

6- In future, please go deeper into the details more detailed data and detailed analysis in subgroups of samples, it is critically required.

Thank you for the recommendation, in further studies, it will be put an effort to go deeper into the details.

7- In spite of declining the ms from publication in Water journal, I have made some remarks which may help authors to publish the submitted ms in the future.

We are very pleased about all the remarks and corrections. They were all analyzed and many changes along the manuscript were done.

8- The structure of the ms is Ok, however the text needs extensive editing, also in terms of language (for example: not “phenomenons” (l.16) but phenomena).

Thank you for accepting the manuscript structure. Many language changes were applied. The paper was submitted for the MDPI services. At the end of this document, it was attached the proof certification that the manuscript has undergone English language editing by MDPI. “Phenomenons” was replaced by “phenomena” in line 17.

9- My main remarks are related to:

10- Abstract(l.15-29) : is not informative enough. Aim of the ms needs to be clarified. Methodological aspects need to be more clear. Results should be reworked and “take-home message” indicated.

As requested it was given more information in the abstract, lines 22-27, lines 29-30 and lines 37-40.

11- Introduction:

Many changes were made in the introduction to fill the requested information, according to the following topics.

12- l. 39 Do really organic loads are contained in urban effluents? Please  be precised.

It was intended to refer that  urban effluents have higher organic loads rather than industrial effluents. Normaly the COD/BOD ratio is higher in industrial effulents (indicating the presence of compounds with low biodegradability) while for Urban effluents the ratio is lower. In order to clarify the reads, it was restructured the phrase in lines (51-52) and also line (54-56).

13- l. 48. What kind of landscape metrics do the authors mean? Please clarify

In line 64 it was added more information about landscape metrics in order to clarify the readers, it was given 3 examples of landscape metrics. It was also added two bibliographical citations to guide any reader that is not familiar with landscape metrics.

14- l. 77. The meaning of “Ecological integrity” needs clarification in the Introduction section. What is the reason to use PLS models to test it?

In lines 82-92 it was added information to clarify the readers about the ecological integrity concept. Resorting to PLS-PM IT WAS established cause-effect relationships between the input variables. For this case variables were pressures in surface waters, the concentration of contaminants and ecological integrity. Since each model contains winter and summer data, and it was used two models, one for short and other for long scale, It was possible to trace how  pressures in surface waters affect SWP and ecological integrity, and how SWP affect ecological integrity for different seasons and accessed scales.

15- l. 84 Hydrographic Region: please give the reference

It was added reference for this nomenclature, line 122

16- l. 87 and 89 How it is possible to consider the land use of the Sabor River Basin as rural while the share of land used by agriculture is 35%?

The classificaon of sabor river basin as rural was rephrased lines (128-130), it was considered as rural due to the low population density and high dependence on livestock and agriculture, as other authors had mentioned.

17- l. 91 average temperature : for which period: month, year, season?

It was added information about the specific period (seasonal), 1˚C in winter and 27.6 in summer, this information was added in lines 128-130.

18- Fig 1  Elevation color gradient is misleading: mountains and highlands are in red, orange and yellow, while lowlands are in green. Watershed divide ( not in legend) should be in red. What are the dashlines in fig 1c.? Moreover  figure caption 1c is not velar: what does it mean: “from this point to the drainage areas”?

It was intended to say from this point (P1) to drainage seccions (dashlines).  The text was changed  in the figure caption. The purpose of figure 1 is to show the location of Sabor river basin in Portugal (figure 1A),  Sabor river basin sampling sites (figure 1B), and Figure C drainage of  sections of sampling site P1. It was not shown the drainage section of 250 meters ( the used in model 1 because it would not be perceptible). It was shown other drainage sections ( 1, 3 , 5 , 7 ,10 and 15 km) as an example. It was added some lines in figure 1 in order to create a “zoom in” concept. As requested, the DEM colour gradient was changed.

19- l. 98: Workflow:  please change into a standard section heading “Materials and methods”

It was changed the title of secion 2.2 to “Materials and Methods”. Since section 2 title was the same, it was also changed into “Methodology”.

20- l. 103-113: Please provide relevant methodology of IPtIs. Macrozoobenthos sampling (2 samples) form P1 are rather ad-hoc and cannot be methodologically justified in this kind of study (to small N).

The equation of IPtIN is represented in appendix C and  as requested it also given the IPtIS equation wich is the same index but applied to the South region of Portugal, lines 119 and 144.

For each site it was measured the index once in winter and another in summer.  It would be better if the index was measured more than once in each season, but due to funding issues it was not possible to do a sampling campaign with more replications. Anyhow, the changes in a  Benthic Invertebrate community require time, since some speciemens lifecycles take months. Nevertheless, we agree that replication of results along all the sampling sites would increase statistical significance , but for funding issues, it was not possible.

21- l. 126 Fig 2  What is the intention to provide the figure? What is its mission?

Figure 2 reveals the general methodology into 3 steps. It was intended to provide this figure to readers to ease the applied methodology. Step 1 was the data collection, SWP, IPtIN and pressures. Step 2 delineation of drainage areas ( short scale, upstream 250 meters), (long scale(entire drainage area).  Step 3 consisted in building two SEM-PLS models, 1 using data from pressures in a upstream distance 250 meters from the SWP and IPTIN sampling sites and in an entire drainage area . The sem pls models have 2 latent variables SWP and Ecological integrity for summer and for winter. Besides the methodology is written along chapter 2 this figure was given to ease the interpretation.

22- l. 137. Unecessary figure  – delete it please.

Figure 1 was changed and edited

23- l. 185: After Terencio please add: et al.

Corrected, line 233

24- Results

25- l.195 Please refer to EU standards for drinking water, as well.

The EU Water Framework Directive 2000/60/EC (WFD) was checked against the Portuguese legislation. It was verified that only for arsenic the threshold limit is stricter.

Portuguese legislation ->50 µg (As)/l

WFD -> 50 µg (As)/l

For this reason in table 2 all the values that were above this threshold were highlighted in orange. It was added a reference to  WFD, lines 236-247.

26- l.96 “more strict” than what?

The sentence was rephrased, lines (242-243)

27- Fig 3.: legend need to be more specified:  What does it mean discharge (kg/km2). Do the author mean N load, maybe? What are the land use conflict classes

Figure 3 and legend were edited, lines (253).The land use conflict classes are the severity of  a land use conflict, varying from 0 to 3. It was added more information in the introduction lines 77-79. It was also added more data in Apeendix A, including table A.1 line 535.

Land use conflics are the difference between the land capability and actual land use. This methodology is applied according to the heavy demands that agricultural areas place in soil and forestry areas conserve it. In terms of land use and land capability is given a score from 1 to 4, agricultural areas (score =1), pastures for livestock production (score =2),  transition areas (forestry or pastures) (score =3) and forest score (score=4). By subtracting the land capability score to the land use score is given the land use conflict. Example, if an area has only capability for forest (score=4) and is occupied by an agriculture field (score =1) the land use conflict class is equal to 3. But if an area has suitability for any land use, each means that is suitable for agriculture (score=1) and is occupied by forestry (score =4) the land use conflict class is not -3 but 0, since in such methodology the classification goes from 0 to 3. This information is resumed in table A.1.  in lines 519-535

28- l. 210-226 No reference to fig 4 (probably). The description of fig 4 should include the interpretation of the drawing.  In its current form, it unnecessarily repeats the content placed on it.

It was made reference to figure 4 in the text and  also it  was added a small interpretation of the figure in the caption, lines 262 and 286.

29- Discussion needs more data-supported conclutions.

It was added more information to support conclusions in topic 4.1, lines 397-479.

Round 2

Reviewer 1 Report

Improved. Please polish the language if possible

Author Response

 Response to revision letter (Round 2) + MDPI English editing certificate

Title: Seasonal and scale effects of anthropogenic pressures on water quality and ecological integrity: A study in the Sabor River Basin (NE Portugal) using PLS-PM

Please polish the language if possible

Many thanks for recognising the improvement of the manuscript. Anyhow, the paper was also read and corrected by an English native speaker, to polish the language (see MDPI certificate in Appendix). All the changes were highlighted in yellow. It was also submitted to the MDPI English editing services. It is indexed the certification, please see below.

Reviewer 2 Report

Thank you for providing the revised version of the manuscript.

After I received the information about the "sampling", I need to state that this was not a representative collection of samples, particularly not for a study with a statistical background. The sampling and the analytics is not documented, and my request to provide this information lead to the result that this practically not exists. If there was done a representative sampling and analytics, ISO norms can be provided. The conclusion on the manuscript is that the methodology is not sound,for that reason it is needed to reject the manuscript. I recommend to re-do the sampling according to scientific principles (sound sampling regime, repeating of the sampling in at least two seasons, several samples), and after this, to re-submit the revised manuscript.

Author Response

Response to revision letter (Round 2) + MDPI English editing certificate

Title: Seasonal and scale effects of anthropogenic pressures on water quality and ecological integrity: A study in the Sabor River Basin (NE Portugal) using PLS-PM

Thank you for providing the revised version of the manuscript. After I received the information about the "sampling", I need to state that this was not a representative collection of samples, particularly not for a study with a statistical background. The sampling and the analytics is not documented, and my request to provide this information lead to the result that this practically not exists. If there was done a representative sampling and analytics, ISO norms can be provided. The conclusion on the manuscript is that the methodology is not sound,for that reason it is needed to reject the manuscript. I recommend to re-do the sampling according to scientific principles (sound sampling regime, repeating of the sampling in at least two seasons, several samples), and after this, to re-submit the revised manuscript.

Thank you in advance for making such a pertinent remark. The usage of ISO norms it is usefull, not only to design international standards but also because they can guide researchers to follow the best scientific practices. The major problem in Portugal is that for environmental research  the founding has been enshorted since. Economic limitations are a massive barrier to research, even still, in projects where such limitations are strictly applied research do their best, to keep science stepping forward. In the scope of the scientific project that this manuscript fits, it was intended to monitor not only Sabor River Basin water quality but also Ave River basin. It was considered by the project stakeholders to  the analysis in a single River basin, and have an improved sampling regime. But there are deep interests to monitor Ave river basin due to it’s historic , and also, continuous contamination. Since the founds had to be split for the monitoring of the two river basins, thus compromising the monitoring of the Sabor river basin.  In order to understand if the measured concentrations are inside regular values of Sabor River basin, it was accessed a Portuguese database SNIRH ( available at https://snirh.apambiente.pt/) where there are measurements of more than 300 parameters across 2000 stations spread across Portugal,  22 of these stations take place in Sabor River Basin. The data of the same SWP parameters were gathered and compared.  The resulting values (for this study) are concordant to the SNIRH analysis, since are inside typical values of Sabor River Basin. Only for potassium, it was not found enough data. Please see Table1 presented in this topic, not in the manuscript. This table was not added to the manuscript since it might confuse the readers about the data sources, it is only being used for revision purposes in order to exemplify that the parameters are in accordance to Sabor River Basin typical values.

            Table1.Maximum, minimum and average values of SWP.

Arsenic

Manganese

pH

Dissolved oxygen

Nitrates

Potassium

µg (As)/l

µg (Mn)/l

mg (O2)/l

mg (NO3)/l

mg (K)/l

Measured
for this
study

Maximum

18.95

551.40

8.78

13.58

15.05

4.05

Minimum

0.03

8.61

6.79

4.03

0.00

0.51

Average

5.39

74.13

7.81

9.33

2.81

1.47

SNIRH

Maximum

24.00

900.00

8.96

14.80

39.10

x

Minimum

1.00

3.20

3.40

2.85

0.01

x

Average

5.64

44.08

7.09

9.12

2.65

x

Reviewer 3 Report

I appreciate that the authors referred to my comments and corrected the text significantly. Their explanations are adequate and satisfying. At the present form, the text is more clear and readable.

Nevertheless, the methodological deficiency in this study  still remains: too small number of samples repeated causes a big error of measurement uncertainty.  However, in light of well presented procedure of the employing PLS-PM model, the manuscript can be published with some minor corrections (abstract and some editing errors).

  The abstract is too extensive, especially after corrections (should be limited to main objectives, methods, results and a core conclusion). As a reader,   I would expect a "take home message" or at least a summary concluding sentence.  

Author Response

Response to revision letter (Round 2) + MDPI English editing certificate

Title: Seasonal and scale effects of anthropogenic pressures on water quality and ecological integrity: A study in the Sabor River Basin (NE Portugal) using PLS-PM

I appreciate that the authors referred to my comments and corrected the text significantly. Their explanations are adequate and satisfying. At the present form, the text is more clear and readable.

We are deeply pleased to hear that the changes are satisfactory. With reviewer's comments the quality of the manuscript has significantly improved.

Nevertheless, the methodological deficiency in this study  still remains: too small number of samples repeated causes a big error of measurement uncertainty.  However, in light of well presented procedure of the employing PLS-PM model, the manuscript can be published with some minor corrections (abstract and some editing errors). The abstract is too extensive, especially after corrections (should be limited to main objectives, methods, results and a core conclusion). As a reader,   I would expect a "take home message" or at least a summary concluding sentence.

Many thanks for these minor revisions. To short the abstract it was applied changes, highlighted in yellow.

Editing errors were corrected, highlighted in yellow along the manuscript.

This manuscript is a resubmission of an earlier submission. The following is a list of the peer review reports and author responses from that submission.

Round 1

Reviewer 1 Report

It is a good work worth publishing. However, some revisions are considered necessary. See attachment

Author Response

In the name of the authors of this paper, we want to thank you for accepting to review this paper. The comments were quite useful to improve the quality of this paper. Some of the comments and questions are answered in this document, the requested changes were made in the paper, highlighted in yellow for the recomended changes, while in blue it was highlighted the English correction changes.

It is a good work worth publishing. However, the following revisions are considered necessary

1.    Line 101-102 “…The second step continued with estimation of an average value for each pressure, within the buffers and the drainage areas…”

Details with regard to this estimation should be provided and references for the calculation of biological index (North Invertebrate Portuguese Index - IPtIN) is needed.

For a better process description, it was added some notes in lines:

114 to 121 where the used arcgis tools are described,

133 to 134 reference to appendix B,

139 to 140 reference to appendix C   

It was added an appendix B and C, where the measurement of swp is described and iptin calculation , respectively.  It was chosen to send the procedures to appendix in order to keep the text structure, and also to keep readers tracked to the simplified methodology description.

2.    Rationale should be provided with regard to the use of both short-scale and long-scale models. Why such a combination can meet the designed aim should be explained. Or, more explanations on Figure 4 is needed. 

In order to clarify the readers about the models construction and algorithm, it was added more information (from line 159 to 181), explaining how the data was used in the sem-pls models. Basically, for the models, data  relative to pressures change in each model (due to the different scales), but the surface water parameters and iptin values are exactly, due to the fact that the sampling points are the same. 

3.    Line 226: “…The short-scale model (M2, lower panel) …”

It should be: The long-scale model (M2, lower panel)

Corrected

4.    How the reliability of the models developed in this study can be evaluated?

For Formative SEM-PLS models, there are three aspects that are mostly tackled by researchers, which are the explained variation by regressors, accessed by R^2, low multicollinearity, accessed by VIF and significance of weights, normally accessed by bootstrapping. When a sem-pls model is applied for prediction purposes this three aspects must be carefully analysed. Is important to have high R^2 squared values to have accurate predictions, VIF bellow 5 in order to do not overestimate effects of variables, and the significance of the weights and effects (total, direct and indirect) should be high (p values lower than 0.05). Since data is normalized in an SEM-PLS algorithm  as higher is the module of weight and or effect, higher is the significance, so another way to access the relative importance is by comparing them inside the model. The presented models were not created to predict, but only to picture and compare season and scale effects. The presented models cannot be used to predict the ecological status, due to the fact that the sample size is low, but they can be used as guidance to create prediction models by choosing the correct season and scale in further studies. Another disadvantage of the presented model is that bootstrapping algorithm cannot be runned, due to the lack of samples. Anyhow, the significance of each variable can be accessed in Figure 5. When the product of the weight by the total effect has the same sign for the same season and or scale, it means that the impact in “SWP” or “ Ecological Integrity” is significant due to the concordance within models. Once again is referred that since data is normalized, so as closer the product is to to zero (presented in figure 5), less is the impact and also less significant is the variable.

5.    Information on water quality measurement methods and instruments will be useful to readers.

Awnsered in comment 1

6.    Although it suggested the fertilizer use is a major pressure in the region, there is no nitrate-related finding. Just wonder if there is any explanation about it.

Is curious that for the impact of nitrates revealed only the decrease of ecological integrity, for winter periods in long scale, and not for short scale models and also for summer in long scale models. It was also checked the Pearson correlation between nitrates and iptin for summer and winter periods, -0.261 and -0.224 respectively. The correlation is negative which indicates a negative impact, but since is so close to zero the isolated effect of nitrates is not enough to justify the locations with lower values of iptin. Besides the impact of nitrates in this study is unclear, because not all the seasons and scales revealed the same results in figure 5, it should be not ignored the impact of nitrates. The concentration of nitrates might be a “numb” threat to water quality. That for other periods might affect water quality. One possible explanation for the unconcluded effect is that the nitrogen cycle Sabor River basin might be balanced, which means that no exceeded nitrogen forms are released in surface waters, for the study period.   

Reviewer 2 Report

It is well organized study riveting on seasonal and scale effects potentially affecting the interactions between anthropogenic pressures, water contamination and ecological integrity in two very different river basins, one urbanized and densely populated and another rural and sparsely populated using partial least squares-path modeling (PLS-PM). This manuscript does an excellent job demonstrating that this statistical model proved efficient in the assessment of intricate relationships, but also alerted for the complex management of multiple

land use watersheds such as the Sabor River Basin. The article is well written, treats an actual problem and provides results that are a challenge to municipal and regional water planners, was well to water management authorities, which are responsible for the safeguard of water quality.

Author Response

In the name of the authors of this paper, we want to thank you for accepting to review this paper. We very much appreciate the nice appreciation on our work

Round 2

Reviewer 1 Report

The revised manuscript can be judged as acceptable.